# Polymorphisms of the *ACSL1* Gene Influence Milk Production Traits and Somatic Cell Score in Chinese Holstein Cows

**DOI:** 10.3390/ani10122282

**Published:** 2020-12-03

**Authors:** Yan Liang, Qisong Gao, Qiang Zhang, Abdelaziz Adam Idriss Arbab, Mingxun Li, Zhangping Yang, Niel A. Karrow, Yongjiang Mao

**Affiliations:** 1Key Laboratory for Animal Genetics, Breeding, Reproduction and Molecular Design of Jiangsu Province, College of Animal Science and Technology, Yangzhou University, Yangzhou 225009, China; MZ120181016@yzu.edu.cn (Y.L.); 18305182715@163.com (Q.G.); yzuzhang785@163.com (Q.Z.); arbabtor@yahoo.com (A.A.I.A.); limingxun@live.com (M.L.); yzp@yzu.edu.cn (Z.Y.); 2Joint International Research Laboratory of Agriculture and Agri-Product Safety of Ministry of Education of China, Yangzhou University, Yangzhou 225009, China; 3Center for Genetic Improvement of Livestock, Department of Animal Biosciences, University of Guelph, Guelph, ON N1G 2W1, Canada; nkarrow@uoguelph.ca

**Keywords:** Holstein cows, SNPs, *ACSL1*, milk production traits

## Abstract

**Simple Summary:**

Milk production traits of cows are important economic indicators of the livestock industry. Many dairy farms strive to improve the quality of their milk. Long-chain acyl-CoA synthetase 1 (*ACSL1*) is a gene related to lipid metabolism. It is widely found in various organisms and can affect fat content and protein content in milk. Single nucleotide polymorphisms (SNP) refers to the polymorphism of DNA sequence caused by a single nucleotide variation at the gene level, which plays a vital function in the genetic study of milk production traits in dairy cows. Our study identified six SNPs of the *ACSL1* gene in Chinese Holstein cows, which were related to milk yield, milk fat content, milk protein content and somatic cell score (SCS) to some extent. In summary, the pleiotropic effects of bovine *ACSL1* for milk production traits were found in this paper, which will provide a reference for Chinese Holstein cow breeding selection and high economic benefits.

**Abstract:**

Improving the quality of milk is a challenge for zootechnicians and dairy farms across the globe. Long-chain acyl-CoA synthetase 1 (*ACSL1*) is a significant member of the long-chain acyl-CoA synthetase gene family. It is widely found in various organisms and influences the lactation performance of cows, including fat percentage, milk protein percentage etc. Our study was aimed to investigate the genetic effects of single nucleotide polymorphisms (SNPs) in *ACSL1* on milk production traits. Twenty Chinese Holstein cows were randomly selected to extract DNA from their blood samples for PCR amplification and sequencing to identify SNPs of the bovine *ACSL1* gene, and six SNPs (5’UTR-g.20523C>G, g.35446C>T, g.35651G>A, g.35827C>T, g.35941G>A and g.51472C>T) were discovered. Then, Holstein cow genotyping (n = 992) was performed by Sequenom MassARRAY based on former SNP information. Associations between SNPs and milk production traits and somatic cell score (SCS) were analyzed by the least-squares method. The results showed that SNP g.35827C>T was in high linkage disequilibrium with g.35941G>A. Significant associations were found between SNPs and test-day milk yield (TDMY), fat content (FC), protein content (PC) and SCS (*p* < 0.05). Among these SNPs, SNP 5’UTR-g.20523C>G showed an extremely significant effect on PC and SCS (*p* < 0.01). The SNP g.35446C>T showed a statistically significant effect on FC, PC, and SCS (*p* < 0.01), and also TDMY (*p* < 0.05). The SNP g.35651G>A had a statistically significant effect on PC (*p* < 0.01). The SNP g.35827C>T showed a highly significant effect on TDMY, FC, and SCS (*p* < 0.01) and significantly influenced PC (*p* < 0.05). Lastly, SNP g.51472C>T was significantly associated with TDMY, FC, and SCS (*p* < 0.05). In summary, the pleiotropic effects of bovine *ACSL1* for milk production traits were found in this paper, but further investigation will be required on the intrinsic correlation to provide a theoretical basis for the research on molecular genetics of milk quality traits of Holstein cows.

## 1. Introduction

Holstein cow is the main breed of dairy cows distributed throughout China. Milk production traits are among the main economic characteristics of Holstein cows, as they are the most direct index to evaluate dairy farms’ management and can directly reflect many problems in the management of dairy cows. The milk production trait of dairy cows is affected by many factors, including genetic, physiological and environmental factors. Some of the key factors directly affect the milk yield level and production potential [1]. Among milk production traits, there was a significant correlation between fat content (FC) and milk yield, protein content (PC), milk urea nitrogen (MUN), and somatic cell count (SCC) [2]. Moreover, mastitis is the most prevalent disease of cows in the world and has led to great economic losses to the dairy industry due to reduced milk production and quality [3]. An indirect strategy of selection for reduced mastitis is based on milk somatic cell score (SCS), which is strongly and positively correlated with clinical mastitis [4].

Recently, significant research progress on the physiology of milk production of Holstein cows has been made [5]. Studies have shown that the detection of single nucleotide polymorphisms (SNPs) and genomes associated with milk production at 305 days could help identify genes associated with milk production traits in cows [6]. For instance, six genes (*ACACA*, *GPAM*, *ACSL1*, *FASN*, *LPIN1* and *ACSL6*) were significantly up-regulated during lactation in Holstein cows [7]. In addition, 20 novel promising genes associated with milk fatty acid traits in Chinese Holstein cows have been identified through genome-wide association analysis; long-chain acyl-CoA synthetase 1 (*ACSL1*) is one of them [8].

*ACSL1* of cattle (*Bos Taurus*), located on chromosome 27, contains 20 exons and 19 introns, with a total length of 64,883 bp. As a member of long-chain acyl-CoA synthetase, *ACSL1* plays a crucial role in the synthesis of triglycerides, phospholipids and cholesterol esters and the oxidation of fatty acids, and is an important candidate gene for dairy quality traits [9,10]. About 98% of milk fat content is comprised of triglycerides and mainly composed of glycerin and long-chain fatty acids [11]. As a key enzyme in fatty acid metabolism, bovine *ACSL1* can produce long-chain fatty acyl-CoA using long-chain fatty acids, adenosine triphosphate, and coenzyme A as substrates [12]. Furthermore, the *ACSL1* gene is a candidate gene for the position and function of fatty acid composition in bovine skeletal muscle [13]; the expression level is the highest in buffalo mammary tissue [14]. Therefore, we hypothesized that the SNPs in *ACSL1* might contribute to variation in milk production traits and SCS. Thus, this study was aimed to investigate potential associations of SNPs in *ACSL1* with milk production traits and SCS in Holstein cows in southern China.

## 2. Materials and Methods

### 2.1. Data and Animal Sample Collection

Phenotypic data were comprised of 12,085 test-day records of 992 Chinese Holstein cows from six different farms located in Jiangsu Province, China. These cows were fed in free-tie stalls, milked three times per day, and fed based on a total mixed ration (TMR). The DC305 software (Valley Ag. software, San Francisco, CA, USA) was used for dairy cow management, including data collection. The data were selected to ensure both reliability and consistency for statistical analyses based on the following criteria: test-day milk yield (TDMY) was between 5 and 60 kg, FC was between 2% and 7%, PC was between 2% and 6%, and SCS was between 0 and 9. Finally, 9076 test-day records were included in this study.

The blood samples were obtained from healthy Chinese Holstein cows randomly selected from the above-mentioned 992 cows on dairy farms in Jiangsu province, China. A standard procedure and the traditional phenol–chloroform procedure were used to extract DNA from blood and dissolved it by TE buffer (Tris +EDTA buffer, used as a dissolving agent to protect nucleic acids from enzymatic degradation) [15]. After ensuring the quality and concentration of DNA, some DNA samples were diluted to 100 ng·µL^−1^ and stored frozen at −20 °C for later use.

### 2.2. SNP Discovery and Genotyping

Primers used for SNP identification within *ACSL1* were designed by Designer software package (Primer Premier 5, PP5, Premier, Ottawa, Canada) according to the sequence provided in GenBank (accession No. NC_037354). PCR temperature gradient was determined by an optimal annealing temperature, (Table 1) and PCR reaction was carried out in a PTC-200 DNA Engine cycler (Bio-Rad, Big Sur, CA, USA). Twenty samples were randomly selected from 992 cow DNA samples to screen for the SNP site and its location. The primer and PCR amplification procedures (total size of the amplification was 8544 bp) were used to amplify the sequence of the selected sites. Finally, 8544 bp of 64,883 bp of the *ACSL1* gene were genotyped. The amplification effect was detected by agarose gel electrophoresis and then sequenced by the Shanghai Sangon Company (Shanghai, China). Three software programs, SeqMan (Invitrogen, Carlsbad, CA, USA), SnapGene Viewer (Invitrogen, Carlsbad, CA, USA), and Vector NTI (Invitrogen, Carlsbad, CA, USA), were then used to analyze the sequences and to find the mutation sites and its location. After discovery of the SNP sites, all samples (992, including the previous 20 samples) were genotype by using the MassARRAy system (Sequenom Inc., San Diego, CA, USA). At the same time, twenty samples were repeated twice (the tester did not know that these twenty samples were repeated) in order to ensure the reliability of SNP analysis results. The results showed the accuracy of SNP genotyping to be 100%. 

### 2.3. Statistical Analyses

The statistical chi-square test was used to determine whether the genotype frequencies deviated from the proportions of Hardy–Weinberg equilibrium (HWE). Conventional population genetics statistical analysis (including gene frequency, genotype frequency, HWE, linkage disequilibrium (LD) analysis, etc.) was performed using genetic online software SHEsis (http://analysis.bio-x.cn/ SHEsis Main.htm) [16,17]. The individual haplotype of each cow were inferred by software Beagle 5.1(Brian L. Browning, Washington, USA) [18]. The least-squares method and general linear model (GLM) of SPSS Ver26.0 (IBM, Armonk, New York, NY, USA) were used to analyze the associations between milk production traits /SCS and genotypes and haplotypes [18,19]. The model was as follows:

Y*_ijklmnop_* = μ + Year*_i_* + Season*_j_* + Parity*_k_* + CS*_l_* + DIM*_m_* + F*_n_* + G*_o_* + e*_ijklmnop_*


In the above model, Y*_ijklmnop_* is the dependent variable (here refers to TDMY, FC, PC and SCS); μ is the overall mean; Year*_i_* is the fixed-effect of the *i*th year (*i* = 2016 to 2018); Season*_j_* is the fixed-effect of the *j*th test season (spring is from March to May, summer is from June to August, autumn is from September to November, and winter is from December to January and February of the following year); Parity*_k_* is the fixed-effect of the *k*th parity (here, the parity of cows is 1 to 3); CS*_l_* is the fixed-effect of the *l*th calving season (here, the division of calving season coincides with the division in test season); DIM*_m_* is the fixed-effect of the *m*th DIM class (DIM is days in milk, here three levels we divided as <100 d, 100 d to 200 d, >200 d); F*_n_* = the fixed-effect of the *n*th farm (*n* = 6, six different farms from Jiangsu Province, China); G*_o_* = the fixed effect of the *o*th genotype or haplotype; e*_ijklmnop_* = the random residual effect. Differences were considered statistically significant at *p* < 0.05. Duncan’s method was used for multiple comparisons among different levels of factors. 

## 3. Results

### 3.1. SNPs within ACSL1

Based on the sequencing of the whole gene, six new SNPs in Holstein *ACLS1* were found. Among them, g.20523C>G was located in 5’UTR; g.35446C>T, g.35651G>A, g.35827C>T, and g.35941G>A were located in intron 2; and g.51472C>T was located in intron 11. Details of the six SNP positions in *ACLS1* are illustrated in Figure 1. The observed genotypic and allelic frequencies of SNPs in *ACSL1* are summarized in Table 2. The number of animals with six specific SNPs are 984, 987, 987, 986, 971 and 984 for g.20523C>G, g.35446C>T, g.35651G>A, g.35827C>T, g.35941G>A and g.51472C>T, respectively (Table 2). The *r*^2^ value was 0.98 between g.35827C>T and g.35941G>A, and the *r*^2^ values between other SNP pairs were all less than 0.4, as shown in Figure 2. Fifteen haplotypes were reconstructed for the SNPs (Table 3), and the frequency of haplotype CCGCGC was the highest (0.31), followed by haplotype CCGTAC (0.268).

### 3.2. Effects of Different Non-Genetical Factors on Milking Traits and SCS of Holstein Cows

The effects of different non-genetical factors on milk production traits and SCS is shown in Table 4. Test year, test season, parity, calving season, days in milk and different farms showed highly significant effects on TDMY, PC and SCS (*p* < 0.01). Test season, calving season, days in milk and different farms showed highly significant effects on FC (*p* < 0.01), and test year had significant effects on FC (*p* < 0.05).

### 3.3. Associations of SNPs in ACSL1 with Milking Traits and SCS

Since SNPs g.35827C>T and g.35941G>A were almost completely linked, we analyzed the association of five SNPs (5’UTR-g.20523C>G, g.35446C>T, g.35651G>A, g.35827C>T and g.51472C>T) with milking traits and SCS. The estimated effects of *ACSL1* on milk production traits and SCS are presented in Table 5. The SNP 5’UTR-g.20523C>G showed a highly significant effect on PC and SCS (*p* < 0.01). The PC of the CC genotype was significantly lower than that of the CG and GG genotype (*p* < 0.05), and the SCS of the CC genotype was significantly higher than that of the CG and GG genotype (*p* < 0.05). With the increase in C>G, PC showed an upward tendency, while SCC showed a downward tendency. The SNP g.35446C>T showed a statistically significant effect on FC, PC and SCS (*p* < 0.01), and had a significant effect on TDMY (*p* < 0.05). Among them, the TDMY and PC of the TT genotype were significantly higher than those of the CC genotype (*p* < 0.05), and the FC and SCS of the TT genotype were significantly lower than those of the CC genotype (*p* < 0.05). Furthermore, with the increase in C>T, TDMY showed an upward tendency. The SNP g.35651G>A showed an extremely significant effect on PC (*p* < 0.01). Specifically, the PC of the GG genotype was significantly higher than that of the GA and AA genotype (*p* < 0.05); PC showed a downward tendency with the increase in G>A. The SNP g.35827C>T showed a highly significant effect on TDMY, FC, and SCS (*p* < 0.01) and significantly affected PC (*p* < 0.05). Moreover, TDMT, FC, PC and SCS all showed downward tendencies with the increase in C>T. For the SNP g.35827C>T, the TDMY and FC of the TT genotype were significantly lower than those of the CC and TC genotypes (*p* < 0.05), and the PC and SCS of the CC genotype were significantly higher than those of the TC and TT genotypes (*p* < 0.05). The SNP g.51472C>T showed significant effects on TDMY, FC and SCS (*p* < 0.05). For the SNP g.51472C>T, the TDMY and FC of the TT genotype were significantly lower than those of the CC and TC genotypes (*p* < 0.05), and TDMY and FC both showed upward tendencies with the increase in C>T.

### 3.4. Associations of Haplotypes for SNPs in ACSL1 with Milking Traits and SCS

The estimated effects of haplotypes for SNPs of *ACSL1* on milk production traits and SCS are presented in Table 6. We retained eleven haplotypes with higher frequencies to analyze and found that different haplotypes of SNPs in *ACSL1* had extremely significant effects on TDMY, FC, PC and SCS (*p* < 0.01). The TDMY of cows with haplotype CCGCGT was 38.91 kg and significantly higher than the TDMY of haplotype CCGCGC, CCGTAC, GGCTGC, CCACGC, CTGCGC, GCCTGC and GCGCAC (*p* < 0.05). The FC of milk with haplotype GCGCAC was 4.15%, which was significantly higher than other haplotypes (*p* < 0.05); the FC of milk with haplotype GTGCGC was 3.49% and significantly lower than haplotype CTGCGC, CCGCGT and GCGCAC. For the PC of cows, the content of haplotype GCGCAC was the highest in all eleven haplotypes (3.38%), and haplotype CCGCAC was the lowest (3.21%). The SCS of milk with haplotype GCGTGC was 1.92, and was significantly lower than haplotype CCGCGC, CCGCAC, CCGTAC, CCGTGC, CCACGC, CTGCGC and CCGCGT (*p* < 0.05). Moreover, the dairy herd improvement (DHI) record number of haplotype CCGCGC was the maximum (4116), and that of haplotype GCGCAC was the minimum (55). In general, cows with haplotype GCGCAC had higher FC and PC, those with haplotype CCGCGT had higher TDMY, and those with haplotype GCGTGC had the lowest SCS.

## 4. Discussion

*ACSL1* is highly expressed in tissues associated with energy metabolism, such as liver, fat, muscle, and breast tissue [13,20]. Hoashi etc. [21] found three polymorphic loci in the second exon (282 bp C/T, 516 bp C/G, 1938 bp T/G) of *ACSL1* in Japanese black cattle. Still, there was no correlation analysis between the polymorphic locus and production traits or milk quality traits. To date, very little information is available about the importance of *ACSL1* in milk production. 

In our study, a total of six novel SNPs were identified in *ACSL1* in Holsteins, and SNPs g.35827C>T and g.35941G>A were in LD. Therefore, five of these SNPs were chosen for further screening to evaluate their potential associations with milk production traits. These SNPs were found to be significantly associated with milk production traits. This research is the first study to examine SNPs’ associations in *ACSL1* with the milk production traits of Holstein cows to the best of our knowledge. Bionaz et al. [22] reported the expression changes of *ACSL1* during lactation in lactating cows and found that the expression of *ACSL1* was upregulated with lactation. In the present study, we found that the SNPs in *ACSL1* were significantly associated with milk production traits and SCS in Holstein cows. The SNPs g.35446C>T, g.35827C>T, and g.51472C>T showed significant associations with TDMY. Furthermore, we found that the SNPs g.35446C>T, g.35827C>T, and g.51472C>T showed significant effects on FC. In bovine mammary tissue, *ACSL1* facilitates the absorption of esterified long-chain fatty acids in fat cells and plays a key role in bovine fat synthesis and fatty acid beta-oxidation [22]. A polymorphism in the yak *ACSL1* gene promoter region also significantly affects FC [23,24]. The above results support the hypothesis that *ACSL1* plays an important role in milk fat synthesis. 

The above studies have shown that the SNPs in *ACSL1* have significant effects on the lactation performance of Holstein cows. The 5’UTR-g.20523C>G is an SNP located in the 5’-nontranslated region, which contains an internal ribosome entry site that can mediate the internal translation initiation of messenger RNA [25]. Thus, the expression of *ACSL1* may be affected by 5’UTR-g.20523C>G, which has an influence on the metabolism of milk fat in Holstein cows and ultimately affects some milk production traits. For the other five SNPs that we found in the intron region, they were all shear sites near the exons upstream and downstream. Mutations at intron splicing sites have been found to cause activation of adjacent covert splicing sites, allowing mature mRNA molecules to retain an intron or snip off an exon, thereby affecting gene expression [26,27]. Additionally, many studies have revealed that introns have positive and negative regulatory effects on gene expression and may have some functions of promoters, and intron mutations in some genes may also cause changes in gene expression levels [22,28]. Thus, although SNPs in the intron region do not cause changes in amino acids, they may affect protein formation by affecting gene splicing. Besides, due to the interaction between environment and genes, natural selection, and other factors (in this experiment, Holstein cows from southern China were selected), lactation performance and SCS of Holstein cows were different. 

## 5. Conclusions

Six SNPs (5’UTR-g.20523C>G, g.35446C>T, g.35651G>A, g.35827C>T, g.35941G>A and g.51472C>T) of *ACSL1* were investigated in Chinese Holstein cows. Associations between these SNPs and TDMY, FC, PC, and SCS were significant. However, these associations will require further investigation concerning their impact on biological and practical relevance because of these SNPs’ potential to alter gene expression.

## Figures and Tables

**Figure 1 animals-10-02282-f001:**
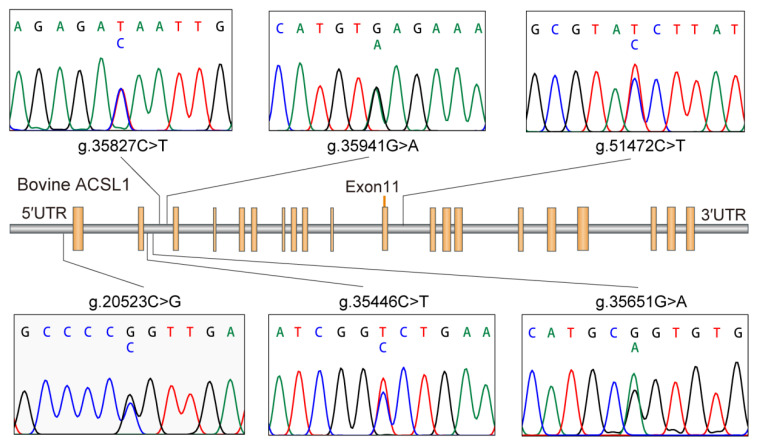
Long-chain acyl-CoA synthetase 1 gene (*ACSL1*) with the localization of the six identified single nucleotide polymorphisms (SNPs).

**Figure 2 animals-10-02282-f002:**
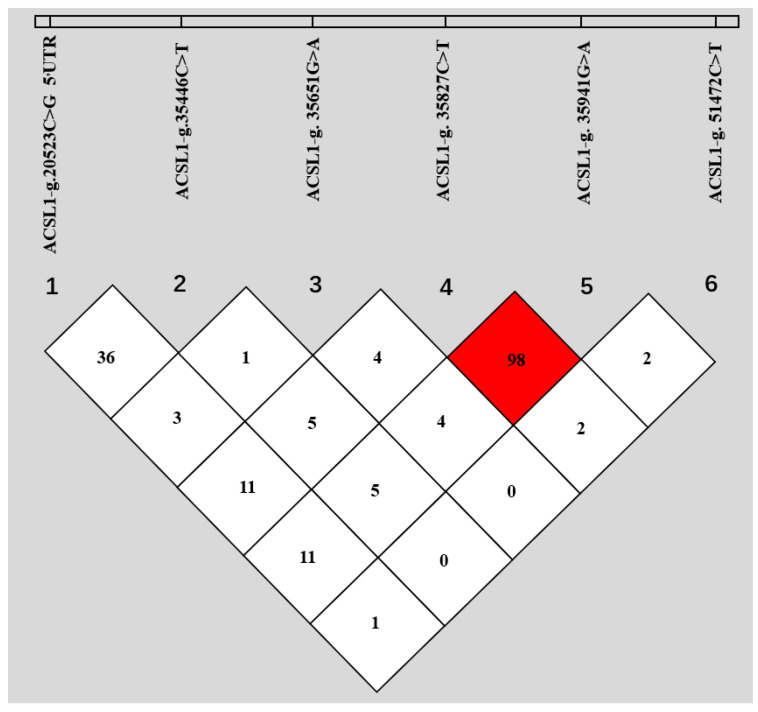
Linkage disequilibrium (LD) among the six SNPs of bovine *ACSL1*. The values in boxes are pairwise SNP correlations (*r*^2^), and the bright red box indicates approximate complete LD (*r*^2^ = 1).

**Table 1 animals-10-02282-t001:** Primer information of PCR amplification for long-chain acyl-CoA synthetase 1 (*ACSL1*) genes.

Primer Name	Primer Sequences (5′→3′)	Size(bp)	Exon	Position	Tm (°C)
P1	F1: AACCCAGCGTGACCTGTTTCACCAG	963	5’UTR + Exon 1	−484~+479	69
	R1: ATGAGCCTCTGCTCCGTGTGTAACG				
P2	F2: GTCCATGCAGCAAACACTCACCC	1070	Exon 2,3 + Intron 2,3	+14,601~+15,648	64
	R2: CAACCTACAGAGGCTCCAGAAA				
P3	F3: ACTGGGCAAGTGTTTTGTTCATTAG	1034	Exon 5,6 + Intron 5,6	+21,364~+22,373	63
	R3: TCGCTCAGTCATGTCCGACTCTTAG				
P4	F4: TCAGCTTGAACTGACTTGATGTGAC	1382	Exon 7–9 + Intron 7–9	+24,637~+25,994	64
	R4: ATAGTCCGGGCCTAACATGATGGTG				
P5	F5: AAGTCCTGCATGGATTTACTTTGTC	485	Exon 10 + Intron 10	+28,070~+28,530	63
	R5: GAACTGCCTACGGGGAAGATGG				
P6	F6: AAGCCACATTCCCTAGTTGCTG	1028	Exon 11 + Intron 11	+30,241~+31,247	60
	R6: GCACTATGAAAGTGGAGGCATC				
P7	F7: TTCCAGTTTCTCCACATCTTCAC	1291	Exon 12,13 + Intron 12,13	+32,505~+33,773	59
	R7: ACACATCCGAAGAAAAGAAGGG				
P8	F8: GGGTCTTTATCCCTTCAGAGGC	1291	Exon 19 + 3’UTR	+42,584~+43,853	60
	R8: CCTATGTCCTAGAATTTTGGCTTG				

**Table 2 animals-10-02282-t002:** Genotypic and allelic frequency, and values of chi-square test significance for SNPs of *ACSL1* genes in Chinese Holstein cows.

SNP Locus	Location	Number	Genotypes	Genotype Frequency	Allele	Allele Frequency	Chi-Square Value for HWE Test
5’UTR-g.20523C>G		581	CC	0.590	C	0.76	4.700
	5’UTR	334	CG	0.339	G	0.24	
		69	GG	0.070			
g.35446C>T		770	CC	0.780	C	0.871	34.399
	Intron 2	180	CT	0.182	T	0.129	
		37	TT	0.037			
g.35651G>A		12	GG	0.012	G	0.107	0.042 *
	Intron 2	188	GA	0.190	A	0.893	
		787	AA	0.797			
g.35827C>T		540	CC	0.548	C	0.729	
	Intron 2	357	CT	0.362	T	0.271	7.00
		89	TT	0.090			
g.35941G>A		525	GG	0.542	G	0.727	
	Intron 2	357	GA	0.369	A	0.273	4.93
		87	AA	0.089			
g.51472C>T		838	CC	0.852	C	0.922	
	Intron 11	139	CT	0.141	T	0.078	0.219
		7	TT	0.007			

*: *p* < 0.05. HWE is Hardy–Weinberg equilibrium.

**Table 3 animals-10-02282-t003:** Haplotype reconstructions for SNP_S_ of the *ACSL1* gene and their frequencies.

Haplotypes	Number	Frequencies
CCGCGC	598	0.31
CCGTAC	517	0.268
GTGCGC	230	0.119
GCGCGC	226	0.117
CCACGC	206	0.107
CCGCGT	116	0.06
CTGCGT	19	0.01
CCGTAC	8	0.004
GCGCGT	5	0.003
CCACAC	2	0.001
CCACGT	1	0.001
CTGCGC	1	0.001
GCGCAC	1	0.001
GTGCAC	1	0.001
GTGCGT	1	0.001
Total	1932	1

**Table 4 animals-10-02282-t004:** Effects of different non-genetical factors on milking traits and somatic cell score (SCS) of Holstein cows.

Factors	Milking Traits and SCS	F Value	Sig
Test year	TDMY	9.63 **	0.00
	FC	4.39 *	0.01
	PC	9.02 **	0.00
	SCS	9.67 **	0.00
Test season	TDMY	69.27 **	0.00
	FC	99.59 **	0.00
	PC	159.50 **	0.00
	SCS	7.26 **	0.00
Parity	TDMY	188.99 **	0.00
	FC	0.52	0.60
	PC	10.80 **	0.00
	SCS	83.49 **	0.00
Calving season	TDMY	23.83 **	0.00
	FC	5.50 **	0.00
	PC	6.94 **	0.00
	SCS	8.40 **	0.00
Days in milk	TDMY	471.99 **	0.00
	FC	97.49 **	0.00
	PC	745.58 **	0.00
	SCS	35.48 **	0.00
Farm	TDMY	49.75 **	0.00
	FC	65.90 **	0.00
	PC	187.06 **	0.00
	SCS	85.29 **	0.00

Analysis of variance adopts the method of the joint hypotheses test (F test). The F value in the result represents a specific value obtained by using the formula of the F test. According to this value, the corresponding *p* value can be obtained by looking up tables or other methods, that is significance (Sig). TDMY is test-day milk yield; FC is fat content; PC is protein content; SCS is somatic cell score; *: *p* < 0.05; **: *p* < 0.01.

**Table 5 animals-10-02282-t005:** Effects of SNPs in *ACSL1* genes on milk production traits and SCS.

SNP Locus	Genotypes	DHI Record Number	TDMY (kg)	FC (%)	PC (%)	SCS
5’UTR-ACSL1-g.20523C>G	CC	5070	34.65 ± 0.15	3.63 ± 0.01	3.22 ± 0.01 ^b^	2.84 ± 0.03 ^a^
	CG	3204	35.40 ± 0.19	3.66 ± 0.02	3.25 ± 0.01 ^a^	2.68 ± 0.04 ^b^
	GG	774	35.70 ± 0.41	3.62 ± 0.03	3.27 ± 0.01^a^	2.57 ± 0.07 ^b^
	Total	9048	34.97 ± 0.11	3.64 ± 0.01	3.24 ± 0.00	2.76 ± 0.02
	F value		2.789	1.734	14.797 **	4.463 **
	Sig		0.06	0.18	0.00	0.01
ACSL1-g.35446C>T	CC	6898	34.82 ± 0.13 ^b^	3.63 ± 0.01 ^a^	3.23 ± 0.00 ^b^	2.79 ± 0.03 ^a^
	CT	1768	35.42 ± 0.26 ^a,b^	3.70 ± 0.02 ^a^	3.24 ± 0.01 ^b^	2.66 ± 0.05 ^a,b^
	TT	410	35.84 ± 0.52 ^a^	3.53 ± 0.05 ^b^	3.28 ± 0.02 ^a^	2.75 ± 0.10 ^b^
	Total	9076	34.97 ± 0.11	3.64 ± 0.01	3.24 ± 0.00	2.76 ± 0.02
	F value		4.220 *	5.002 **	6.279 **	8.532 **
	Sig		0.02	0.01	0.00	0.00
ACSL1-g.35651G>A	AA	71	35.01 ± 1.43	3.75 ± 0.11	3.22 ± 0.04 ^b^	2.62 ± 0.25
	GA	1660	34.45 ± 0.27	3.64 ± 0.02	3.22 ± 0.01 ^b^	2.78 ± 0.05
	GG	7338	35.08 ± 0.13	3.64 ± 0.01	3.24 ± 0.00 ^a^	2.76 ± 0.02
	Total	9069	34.97 ± 0.11	3.64 ± 0.01	3.24 ± 0.00	2.76 ± 0.02
	F value		1.262	0.704	7.016 **	0.007
	Sig		0.28	0.50	0.00	0.99
ACSL1-g.35827C>T	CC	5143	35.19 ± 0.15 ^a^	3.67 ± 0.01 ^a^	3.25 ± 0.01 ^a^	2.83 ± 0.03 ^a^
	TC	3180	35.03 ± 0.19 ^a^	3.62 ± 0.02 ^a^	3.22 ± 0.01 ^b^	2.67 ± 0.04 ^b^
	TT	740	33.24 ± 0.38 ^b^	3.56 ± 0.03 ^b^	3.22 ± 0.01 ^b^	2.67 ± 0.08 ^b^
	Total	9063	34.97 ± 0.11	3.64 ± 0.01	3.24 ± 0.00	2.76 ± 0.02
	F value		4.869 **	5.301 **	3.984 *	14.045 **
	Sig		0.01	0.01	0.02	0.00
ACSL1-g.51472C>T	CC	7592	35.04 ± 0.13 ^b^	3.63 ± 0.01 ^b^	3.23 ± 0.00	2.78 ± 0.02 ^a^
	TC	1376	34.56 ± 0.28 ^b^	3.70 ± 0.02 ^b^	3.27 ± 0.01	2.66 ± 0.06 ^b^
	TT	76	38.69 ± 1.23 ^a^	3.92 ± 0.10 ^a^	3.26 ± 0.04	2.81 ± 0.22 ^a,b^
	Total	9044	34.97 ± 0.11	3.64 ± 0.01	3.24 ± 0.00	2.76 ± 0.02
	F value		3.169 *	4.359 *	0.371	3.412 *
	Sig		0.04	0.01	0.69	0.03

Analysis of variance adopts the method of the F test. The F value in the result represents a specific value obtained by using the formula of the F test. According to this value, the corresponding *p* value can be obtained by looking up tables or other methods, that is, Sig. DHI is dairy herd improvement; TDMY is test-day milk yield; FC is fat content; PC is protein content; SCS is somatic cell score; *: *p* < 0.05; **: *p* < 0.01; ^a,b^ differences in the same column are significant at *p* < 0.05.

**Table 6 animals-10-02282-t006:** Effects of haplotypes for SNPs on milk production traits and SCS.

Haplotypes	DHI Records Number	TDMY (kg)	FC (%)	PC (%)	SCS
CCGCGC	4116	34.89 ± 0.17 ^b,c^	3.64 ± 0.01 ^b,c^	3.23 ± 0.01 ^b^	2.91 ± 0.03 ^a,b^
GCGCGC	1335	35.56 ± 0.30 ^a,b,c^	3.69 ± 0.02 ^b,c^	3.26 ± 0.01 ^b^	2.48 ± 0.05 ^a,b,c,d^
CCGCAC	1182	35.63 ± 0.30 ^a,b,c^	3.59 ± 0.03 ^b,c^	3.21 ± 0.01 ^b^	2.69 ± 0.06 ^a,b,c^
CCGTAC	1133	34.05 ± 0.32 ^b,c^	3.59 ± 0.03 ^b,c^	3.22 ± 0.01 ^b^	2.73 ± 0.06 ^a,b,c^
GTGCGC	355	36.10 ± 0.56 ^a,b^	3.49 ± 0.05 ^c^	3.29 ± 0.02 ^ab^	2.58 ± 0.10 ^a,b,c,d^
CCGTGC	354	33.77 ± 0.51 ^b,c^	3.67 ± 0.05 ^b,c^	3.24 ± 0.02 ^b^	2.65 ± 0.12 ^a,b,c^
CCACGC	225	34.12 ± 0.73 ^b,c^	3.71 ± 0.06 ^b,c^	3.26 ± 0.02 ^b^	3.00 ± 0.17 ^a^
CTGCGC	198	34.27 ± 0.71 ^b,c^	3.88 ± 0.06 ^b^	3.24 ± 0.03 ^b^	3.07 ± 0.16 ^a^
CCGCGT	85	38.91 ± 1.11 ^a^	3.81 ± 0.10 ^b^	3.26 ± 0.04 ^b^	2.64 ± 0.20 ^a,b,c^
GCGTGC	61	32.31 ± 1.49 ^c^	3.61 ± 0.10 ^b,c^	3.30 ± 0.03 ^a,b^	1.92 ± 0.20 ^d^
GCGCAC	55	33.41 ± 1.15 ^b,c^	4.15 ± 0.10 ^a^	3.38 ± 0.05 ^a^	2.17 ± 0.27 ^c,d^
Total	9099	34.97 ± 0.11	3.64 ± 0.01	3.24 ± 0.00	2.76 ± 0.02
F value		3.979 **	5.136 **	3.201 **	6.749 **
Sig		0.00	0.00	0.00	0.00

Analysis of variance adopts the method of the F test. The F value in the result represents a specific value obtained by using the formula of the F test. According to this value, the corresponding *p* value can be obtained by looking up tables or other methods, that is, Sig. TDMY is test-day milk yield; FC is fat content; PC is protein content; SCS is somatic cell score; **: *p* < 0.01; ^a,b,c,d^ differences in the same column are significant at *p* < 0.05.

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
