# Peer review of "Polymorphisms of the *ACSL1* Gene Influence Milk Production Traits and Somatic Cell Score in Chinese Holstein Cows"

_animals, 2020, doi:10.3390/ani10122282_

Round 1
Reviewer 1 Report
The paper was bettered, the authors followed the comments. Only some small revisions are needed, but the article will be of interest for Animals.
- Rr. 74, 78, ACSL1 as enzyme must be written in normal type.
- MM section, chapter 2.2, r. 110: ….were genotyped by using….
- Please, write in this section a short notice that 8544bp of 64883bp of the ACSL1 gene were genotyped.
- Chapter 2.3, r. 121, cow not cows. R. 134 genotype or haplotype, not genotypes, haplotypes.
- Tables 2 and 5, genotypes in italic.
- In chapter 3.3, the MUN is discussed, but it is not in the Table 5.
- 190, ….increase of G>A. “space” SNP….
- Better the legend below the Table 5:
Add explanation of * and **.
a,bdifferences in the same column are significant at P<0.05.
- Add legend below the Table 6.
- Section 3.6: r. 204 ACSL1 in italic; r. 207 delete Among; r. 209 FC not PC.
- Tables 4 and 6 are well computed and commented, thumbs up!
- R. 258 5 “point” Conclusions
Author Response
All of my original comments have been addressed and have minor comments on the revised manuscript:
- Rr. 74, 78, ACSL1 as enzyme must be written in normal type.
Response: Thanks for your reminder and we have already modified the type of ACSL1, in line74 and 78 as following “long-chain acyl-CoA synthetase 1 (ACSL1) is one of them [8]……As a member of long-chain acyl-CoA synthetase, ACSL1 plays a crucial role”
2. MM section, chapter 2.2, r. 110: ….were genotyped by using….; Please, write in this section a short notice that 8544bp of 64883bp of the ACSL1 gene were genotyped.
Response: Thanks for your comment. “detected” have been deleted (line 113) and added "8544bp of 64883bp of the ACSL1 gene were genotyped" in this section. As is shown in lines 106-109, as following “The primer and PCR amplification procedures (total size of the amplification was 8544 bp) were used to amplify the sequence of the selected sites. Finally, 8544 bp of 64883 bp of the ACSL1 gene were genotyped.”
- Chapter 2.3, r. 121, cow not cows. R. 134 genotype or haplotype, not genotypes, haplotypes.
Response: Thanks for your comment. Some formal errors have been fixed. “cows” and “genotypes or haplotypes” were changed to “cow” and “genotype or haplotype”. They were shown in line124 and line 137.
- Tables 2 and 5, genotypes in italic.
Response: We have corrected the incorrect font style here, as is shown in Tables 2 and 5.
- In chapter 3.3, the MUN is discussed, but it is not in the Table 5.
Response: Thanks for your comment. We have deleted discussion on MUN in chapter 3.3.
- Line 190, ….increase of G>A. “space” SNP….
Response: Thanks. We have added “space” there. As is shown in line 193, as following “……with the increase of G>A. SNP g.35827C>T showed……”
- Legend below the Tables: add explanation of * and **; a,b differences in the same column are significant at P<0.05; add legend below the Table 6.
Response: Thanks. We have added legend below the Table 5 and Table 6.
- Section 3.6: r. 204 ACSL1 in italic; r. 207 delete Among; r. 209 FC not PC.
Response: Thanks. We have corrected the incorrect font style (line 212) and deleted “Among” (line 215). Besides, “PC” was changed to “FC” shown in line 217.
- Tables 4 and 6 are well computed and commented, thumbs up!
Response: Thank you very much, and I will try our best to further improve our computed and commented, thanks.
- R. 258 5 “point” Conclusions
Response: Thanks for your reminder and we have added “point” between “5” and “ Conclusions”. As is shown in line 285, as following “5. Conclusions”.
Reviewer 2 Report
The work is relevant and the modifications required by the reviewers were considered. I believe the data will support further investigations concerning the impact of gene expression on biological and practical relevance.
Author Response
The work is relevant and the modifications required by the reviewers were considered. I believe the data will support further investigations concerning the impact of gene expression on biological and practical relevance.
Response: Thank you very much and I will try our best to further improve our manuscript, thanks.
Reviewer 3 Report
Dear Authors,
All my previous suggestions and suggestions from other Reviewers` have been introduced into the manuscript and now the introduction is enough described to understand the subject of study. Material and methods are properly described and consist of details, which are necessary for this type of article. The result and discussion are now properly, clearly described, and much more improved after other reviewers' comments. I am content with the new version of this manuscript.
kind regards
Author Response
All my previous suggestions and suggestions from other Reviewers` have been introduced into the manuscript and now the introduction is enough described to understand the subject of study. Material and methods are properly described and consist of details, which are necessary for this type of article. The result and discussion are now properly, clearly described, and much more improved after other reviewers' comments. I am content with the new version of this manuscript.
Response: Thank you very much and I will try our best to further improve our manuscript, thanks.
This manuscript is a resubmission of an earlier submission. The following is a list of the peer review reports and author responses from that submission.
Round 1
Reviewer 1 Report
The paper deals with polymorphisms in ACSL1 gene. It brings new knowledge of interest, but revisions are needed. There are some imperfections mainly in the presentation of results.
- The 1st paragraph of introduction is not well written, re-formulate it. Write 2-3 sentences on the importance of milk production, e.g.
- 2nd paragraph, describe better the gene. Give BTA, number of exons, introns, length, etc.
- Results section, rr. 134-151, the authors just repeat the content of Table 4. Please, try to describe tendencies etc.
- Would you like to test the differences among haplotypes, condensed genotypes and the like?
- The effects of season, farm, DIM were significant?
- Discussion section, rr. 177-179, delete sentences In goat….(Zhao et al., 2019).
- 183-185, delete In our…SCS.
- 185-193, the text has nothing to do with the topic of the article. Delete In the…and SCC.
- 194-209 are OK.
- Conclusions section, r. 211, delete For…time.
- Generally, the style is a bit clumsy.
Formal errors:
- Title r. 2, ACSL1 in italic.
- Abstract r. 21, etc., not et al.
- Abstract r. 30 in the end, PC, not FC.
- Rr. 47, 61, explain abbreviations.
- MM section, r. 99, explain the abbreviation.
- MM section, rr. 103-104, spring etc. is the same as for calving season?
- Table 2, write correct chi-square, not X
- Table 2 and 4, the colons in legend are according to journal style?
- Table 4, legend, explain Sig.
Finally, after some corrections the article will be of interest for Animals.
Author Response
Responds to the reviewer s' comments:
(The font in black is comments, in green is authors’ responses)
All of my original comments have been addressed and have minor comments on the revised manuscript:
- The 1st paragraph of introduction is not well written, re-formulate it. Write 2-3 sentences on the importance of milk production, e.g
Response: Thanks for your reminder and, we added a description of the importance of milk production in the 1st paragraph of the introduction. They were shown in lines 58-61.
- 2nd paragraph, describe better the gene. Give BTA, number of exons, introns, length, etc.
Response: We have added genetic information about ACSL1. As is shown in lines 75-76: “ACSL1 of Bos taurus, located on chromosome 27, contains 20 exons and 19 introns, with a total length of 64 883 bp”.
- Results section, rr. 134-151, the authors just repeat the content of Table 4. Please, try to describe tendencies etc.
Response: We have added the description of tendencies for Table 4. They were shown in line180-202 as following “With the increase of C>G, PC showed an upward tendency, while SCC showed a downward tendency.……TDMY and FC all showed upward tendencies with the increase of C>T.”
- Would you like to test the differences among haplotypes, condensed genotypes and the like?
Response: We have added the haplotype analysis on Table 6, added the related description on “3.4 Associations of haplotypes for SNPs in ACSL1 with milking traits and SCS”.
- The effects of season, farm, DIM were significant?
Response: Thanks for your comment. The effects of season, farm, DIM were significant on milking traits and SCS. We have added it. Table 4 to show the effects of different factors on milking traits and SCS.
- Discussion section, rr. 177-179, delete sentences “In goat….”
Response: We have deleted this sentence in the discussion.
- Line 183-185, delete “In our…SCS”, line 185-193, the text has nothing to do with the topic of the article. Delete “In the…and SCC”.
Response: We have deleted these sentences which irrelevant to the topic of the article in the discussion section.
- Conclusions section, r. 211, delete “For…time”.
Response: We have deleted “For…time” in the conclusions section.
- Generally, the style is a bit clumsy.
Response: Thank you very much, and I have generally done the best in setting the style to improve our manuscript further thanks.
- Some formal errors: title r. 2, ACSL1 in italic
Response: Thanks. We have corrected the incorrect font style here, as is shown in the title.
- Abstract r. 21, etc., not et al.
Response: Thanks for your constructive comment, and we have corrected the written error here, as is shown in the abstract r. 32.
- Abstract r. 30 in the end, PC, not FC.
Response: Thanks for your comment. We mistyped here, and we have corrected the abbreviation from “FC” to “PC”, as is shown in abstract r. 41.
- Rr. 47, 61, 99, explain abbreviations.
Response: We had added an explanation when the abbreviation firstly appeared in the text.
- MM section, rr. 103-104, spring etc. is the same as for calving season?
Response: Thanks for your reminder. Firstly, they are different in meaning. The meaning of “Season” such as spring etc., is “test season”, it represented the effects of the season on milk production traits when we performed the DHI. However, “calving season” means the season when cows are calving in this lactation. Moreover, the seasons were divided into spring, summer, autumn and winter according to the local climatic conditions in Jiangsu Province (subtropical monsoon climate), and “test season” and “calving season” were divided in the same way. Maybe our statement is a little misleading, and we change the sentence, as is shown in line 133.
- Table 2, write correct chi-square, not X
Response: We have corrected the word from “X” to “chi-square”, as is shown in Table 2.
- Table 2 and 4, the colons in legend are according to journal style?
Response: Thanks for your constructive comment. We have checked the format of colons in legend and made corrections according to journal style.
- Table 4, legend, explain Sig.
Response: We have added the explanation of “Sig” in Table 4. The revised is shown as follows: “Analysis of variance adopts the method of F test. The F value in the result represents a specific value obtained by using the Formula of F test. According to this value, the corresponding P value can be obtained by looking up tables or other methods, that is, Sig.”.
Reviewer 2 Report
The data obtained from the individuals was sufficiently enough to identify the six SNPs of ACSL1 gene in Chinese Holstein cows. The genetic importance of the chosen protein will provide valuable information for economic benefit.
I identify only one issue. The quality of the figure 2 should be improved, and the table 2 could be presented in a better way (the space between columns is not enough).
Author Response
Responds to the reviewer s' comments:
(The font in black is comments, in green is authors’ responses)
- I identify only one issue. The quality of the figure 2 should be improved, and the table 2 could be presented in a better way (the space between columns is not enough).
Response: Thank you very much. We have improved the quality and format of Figure 2 and Table 2, and I will try our best to further improve our manuscript.
Reviewer 3 Report
The authors describe the influence of SNPs in the ACSL1 gene on milk performance in Chinese Holstein cows.
The introduction is not sufficient. Please add some more information about conducted different research on Chinese Holstein cow. eg. : https://www.sciencedirect.com/science/article/pii/S0022030219306009
https://onlinelibrary.wiley.com/doi/full/10.1111/age.12792
https://journals.plos.org/plosone/article?id=10.1371/journal.pone.0096186
line 31- it is a missing reference?
line 42- and 45 PC, MUN- appear first time in the text- please explain the abbreviation.
I am a little bit confused because if I good understand you collected blood samples from 20 randomly selected Holstein cows and sequenced the whole ACSL1 gene. Based on results from only 20 animals, you concluded that SNPs in ACSL1 could influence milk production traits? You used a MassARRAy system to genotype detected SNPs in those 20 animals or in more? Please clarify this in the text. This is very important information.
Table 2. Please clarify " Number" - Number of what? animals with mutations?
If it is the number of animals why are a different number of animals between specific SNPs?
Line 124- please add information about the number of animals with specific SNPs.
LIne 96- You write about using MassARRAy in genotyping detected 2 loci. Could you add information about the numbers of animals used to genotype
Author Response
Responds to the reviewer s' comments:
(The font in black is comments, in green is authors’ responses)
- The introduction is not sufficient. Please add some more information about conducted different research on Chinese Holstein cow.
Response: Thank you very much and we have added some more information about conducted different research on Chinese Holstein cow as shown in line 67-74.
- The line 31- it is a missing reference?
Response: Thanks for your comment. We mistyped here, and we have corrected the abbreviation from “FC” to “PC”, as is shown in abstract r. 41.
- The line 42- and 45 PC, MUN- appear first time in the text- please explain the abbreviation.
Response: Thanks for your constructive comment and we have added an explanation when the abbreviation first appears.
- I am a little bit confused because if I good understand you collected blood samples from 20 randomly selected Holstein cows and sequenced the whole ACSL1 gene. Based on results from only 20 animals, you concluded that SNPs in ACSL1 could influence milk production traits? You used a MassARRAy system to genotype detected SNPs in those 20 animals or in more? Please clarify this in the text. This is very important information.
Response: Thanks for your comment. The 20 randomly selected samples in the early stage were only used to detect their SNP. Based on this results, increase the sample size for SNP analysis (the sample size is 992). Besides, for SNP sites previously discovered, all these samples (992) were detected by using the MassARRAy system (including the previous 20 samples). Finally, the association analysis were based on the all samples (992 cows) and their milking traits.
- Table 2. Please clarify " Number" - Number of what? animals with mutations?
Response: Thanks for your comment. "Number" is the number of cows, which is number of the animals.
- If it is the number of animals why are a different number of animals between specific SNPs?
Response: Thanks for your comment. Of course, different SNP loci in the population have different genotype distribution unless they are completely linked (such as r2 =1).
- Line 124- please add information about the number of animals with specific SNPs.
Response: Thanks for your comment. We have added information about the number of animals with specific SNPs on Results with “3.1 SNPs within ACSL1”. Line 145-147.
- Line 96- You write about using MassARRAy in genotyping detected 2 loci. Could you add information about the numbers of animals used to genotype.
Response: Thanks for your comment. In fact, all the six SNPs sites were detected by the MassARRAy system. This may be due to ambiguity caused by our improper expression, and we have revised it in the manuscript (line-103, 2.2 SNP discovery and genotyping). Besides, only because of the equipment and technique, the detection rate for different SNP sites are different, resulting in small differences in the number of detected animals. We have listed the numbers of animals of each genotype for each SNP in Table 2.